# Effect of a Self-Assembly Peptide on Surface Roughness and Hardness of Bleached Enamel

**DOI:** 10.3390/jfb13020079

**Published:** 2022-06-13

**Authors:** Gabriela de A. P. Magalhães, May Anny A Fraga, Isaac J. de Souza Araújo, Rafael R. Pacheco, Américo B. Correr, Regina M. Puppin-Rontani

**Affiliations:** 1Department of Restorative Dentistry, Dental Materials Division, Piracicaba Dental School, UNICAMP, State University of Campinas, Piracicaba 13414903, Brazil; gabrielaapmag@gmail.com (G.d.A.P.M.); mayannyfraga@hotmail.com (M.A.A.F.); isaacjsouzaa@gmail.com (I.J.d.S.A.); acorrer@unicamp.br (A.B.C.); 2Department of Health Sciences and Pediatric Dentistry, Pediatric Division, Piracicaba Dental School, UNICAMP, State University of Campinas, Piracicaba 13414903, Brazil; 3Department of Restorative Dentistry, University of Detroit Mercy School of Dentistry, Detroit, 48208 MI, USA; pachecrr@udmercy.edu

**Keywords:** tooth bleaching, tooth remineralization, self-assembly peptide, sodium fluoride, hardness, roughness

## Abstract

After bleaching, enamel surfaces are damaged, contributing to erosion and tooth sensitivity. Although fluoride is used after bleaching to try and revert alterations, it is not capable of repairing tooth structure. This study compared the effect of a self-assembly peptide (P_11_-4), with and without fluoride, and sodium fluoride (NaF 2%) on the Knoop microhardness (KHN) and surface roughness (Ra (μm)) of bleached enamel with an in-office bleaching regimen. Enamel blocks of bovine teeth (5 × 5 × 2 mm) with standardized surface hardness were bleached with 35% carbamide peroxide, following the manufacturer’s instructions. The teeth were randomly divided into the following groups (*n* = 7) according to post-bleaching treatment: no treatment (negative control) (C-); 2% NaF (NaF); Curodont™ Repair (Repair); and Curodont™ Protect (Protect). Specimens were stored in artificial saliva at 37 °C. To evaluate the effect of the post-bleaching treatments, KHN and Ra were measured before bleaching (baseline) and 24 h and 7 days after bleaching. Data were submitted to repeated measures ANOVA and Bonferroni tests (α = 0.05). There were significant interactions between the study factors (*p* = 0.001). After 7 days, Repair (572.50 ± 79.04) and Protect (583.00 ± 74.76) specimens showed increased surface KHN, with values higher than the NaF (465.50 ± 41.50) and C- (475.22 ± 58.95) baseline values. There was no significant difference in KHN at 24 h among groups (*p* = 0.587). At 24 h after bleaching, Repair was significantly different from all groups (*p* < 0.05). Repair showed the lowest Ra (μm) values (0.133 ± 0.035). After seven days, there was no significant difference in Ra values among groups when compared to the baseline. The use of P_11_-4-based materials after bleaching resulted in the fastest recovery to baseline enamel properties.

## 1. Introduction

Dental bleaching is one of the most common aesthetic treatments performed in dentistry and it has changed the way people perceive their own smiles. The number of bleaching procedures has increased due to their relatively quick results, non-invasive techniques, ease of use, and possible reduction in the number of restorative procedures [1,2]. The procedure involves the diffusion of bleaching agents (usually hydrogen or carbamide peroxide) through the tooth structure, followed by the interaction of dissociated peroxide molecules and organic chromophores [3]. Bleaching agents react with water from the dental structure, releasing free radicals as reactive oxygen species (ROS) that break double bonds into single bonds, which are easily broken by chemical oxidation. Large molecules (chromophores) are broken into smaller molecules that absorb less light, hence reducing discoloration [4].

Despite the favorable color change, side effects caused by bleaching with peroxide, such as demineralization, erosion, and tooth sensitivity, have been described [5]. These alterations to enamel surfaces can be exacerbated by an acidic diet [6], increasing surface porosity, surface roughness, and erosion [7]. Due to the low pH of bleaching agents [8], enamel softening may occur. Translucency changes in the enamel can be observed due to deproteinization, demineralization, and oxidation of the enamel organic matrix. This can also increase mineral loss and consequently lead to micromorphological alterations [6,7,8,9,10,11]. It has been demonstrated that surface changes related to peroxide type and concentration, as well as the protocol used, might be recovered by contact with saliva or remineralizing toothpastes. However, the process takes time and may cause discomfort to patients [7,11].

Multiple approaches have been investigated to minimize the erosive effects of bleaching procedures using sodium fluoride-based (NaF) materials and casein phospho-peptide/amorphous calcium phosphate (CPP/ACP) [12,13]. Fluoride was the first and most commonly used compound in post-bleaching treatment due to its ability to promote calcium and fluoride deposition on enamel surfaces [14]. Fluoride ions are incorporated into demineralized areas forming calcium fluoride. These ions can also substitute hydroxyl groups on enamel apatite, resulting in the formation of fluorapatite [14], which significantly reduces mineral loss and recovers enamel microhardness [15]. Nevertheless, fluoride remains available on the enamel surface only, without promoting in-depth remineralization, and can easily be lost after toothbrushing [14].

Biomimetic materials might be a good option to stop the erosion process and consequently preserve tooth structure while avoiding other problems, such as hypersensitivity. Recently, a self-assembling peptide (P_11_-4) that is mimetic to enamel proteins has been developed and introduced on the market for enamel regeneration. This peptide can be found in Curodont™ Repair, a commercial product that is indicated for initial caries lesions, and in Curodont™ Protect, a professional remineralizer gel indicated during orthodontic treatments and after bleaching. This peptide (Figure 1) is pH-responsive and, in slightly acidic conditions, forms a three-dimensional matrix that can support apatite-like crystal growth [16]. Furthermore, P_11_-4 mimics the mechanisms involved in the early stages of the enamel biomineralization process (Figure 2), when crystallization occurs around the enamel organic matrix. While the potential of fluoride to protect enamel is restricted to the outer 30 μm [17], P_11_-4 application has been reported to recover 52 to 103 μm in depth of enamel early caries lesions 14 days after application [16]. Furthermore, previous studies have described substantial mineral gain on carious lesions in human enamel treated with self-assembling peptides [18,19,20,21,22,23].

Other peptide-based self-assembling systems have been studied for biomedical applications, such as PEG8-(FY)3. This is a hybrid polymer–peptide conjugate that can self-assemble into a self-supporting soft hydrogel over dry and wet surfaces. In dentistry, its use could be important, since it acts as a scaffold and supports cell growth [24]. An arginine-containing peptide hydrogel, enriched with hydroxyapatite, seems to be promising for tooth mineralization. This multicomponent peptide-based hydrogel is composed of fluorenyl-9-methoxycarbonyl diphenylalanine (FmocFF), which provides rigidity and stability to the hydrogel, and Fmoc-arginine (FmocR), which mediates a high affinity to hydroxyapatite (HAP) due to an arginine moiety [25].

Apart from mineral content change, dental bleaching can promote a significant increase in the proteolytic activity of cathepsin B and matrix metalloproteinase, suggesting a dynamic modification within the tooth structure [26]. In this regard, P_11_-4 has demonstrated potential to reduce proteolytic activity [27], suggesting that its protective effect goes beyond mineral aggregation. Furthermore, P_11_-4 also shows a protective effect on demineralization when applied on eroded enamel surfaces, being suggested for erosion treatment [28] and prevention [28,29]. However, there is sparse information in the literature about the effects of P_11_-4 on recovering enamel properties after bleaching procedures.

Thus, the aim of this preliminary study was to evaluate the efficacy of different materials containing a self-assembling peptide used in a post-bleaching procedure for recovering the surface properties of enamel. The tested hypotheses were: (1). that P_11_-4-based materials are more efficient than NaF for recovering the surface roughness and microhardness of bleached enamel; and (2). that P_11_-4-based materials form more stable interactions with enamel surfaces than NaF over time.

## 2. Materials and Methods

This study was based on a single randomized model consisting of thirty-one (31) bovine teeth assigned into four (4) groups (*n* = 7) and an additional three sound samples (*n* = 3) for scanning electron microscopy. Sample calculation was accomplished based on a previously conducted pilot study considering α = 0.05 and β = 0.20. Thus, seven specimens per group was deemed adequate. Groups were divided according to enamel surface treatment: a negative control (bleached enamel) (C-); bleached enamel treated with NaF 2% (NaF); bleached enamel treated with Curodont™ Repair (Repair); and bleached enamel treated with Curodont™ Protect (Protect). For qualitative surface comparison, SEM micrographs were obtained with sound (non-bleached) bovine enamel (*n* = 3). Table 1 describes the protocols applied for all groups; Table 2 presents the materials’ compositions and application regimes. Hardness and surface roughness values obtained for specimens during baseline (before enamel bleaching/sound enamel) were considered as a positive control set. Both tests (microhardness and surface roughness) were evaluated for all specimens. The sample distribution and study design can be observed in Figure 3.

Seventy (70) healthy bovine incisor teeth were collected, cleaned, and stored in 0.1% thymol solution at 4 °C for a maximum of two months after extraction. The teeth were then randomized, using a coin-toss method, and distributed into groups. Buccal enamel surfaces were flattened with 400-grit sandpaper under refrigeration until parallel surfaces were obtained with no dentin exposure. After that, the teeth were submitted to Knoop microhardness analysis, and twenty-eight (28) specimens were selected, with average microhardness values ±20%. Specimens were wax-fixed on an acrylic resin platform and sectioned in the buccal–lingual direction using a low-speed water-cooled diamond saw (Isomet 1000, Buehler Ltd., Lake Bluff, IL, USA). The final square-shaped blocks (5 × 5 × 2 mm) were randomly distributed among groups and embedded in self-curing acrylic resin cylindrical stubs, leaving the enamel surfaces exposed. The enamel surfaces were finished and polished using #400-, #600-, and #1,200-grit SiC sandpapers. The specimens were rinsed with deionized water for 30 s for each sandpaper exchange. The microhardness and surface roughness of enamel blocks were analyzed before bleaching procedures (baseline).

Dental bleaching was performed as follows: 35% Carbamide Peroxide (Whiteness HP maxx, FGM, Joinville, SC, Brazil) was mixed according to the manufacturer’s instructions (3 drops of thickener for 1 drop of peroxide) to obtain a homogeneous gel mixture. The mixture was then applied on the bovine enamel surfaces for 15 min. Excess of bleaching gel was removed with gauze. Specimens were rinsed with deionized water for 30 s, three (3) times, and dried using absorbing paper [30].

The C- group was stored in artificial saliva immediately after bleaching. For the NaF group, 2% NaF was left for 1 min on the enamel surface. After that, excess was removed with absorbent paper, and 50 µL of Ca^2+^ and PO_4_ supersaturated solution was applied for 1 min. Excess was removed with gauze, surfaces were rinsed with deionized water for 10 s, and the specimens were stored in artificial saliva (1.5 mM CaCl_2_, 0.9 mM KH_2_PO_4_, 130 mM KCl, and 20 mM Hepes, pH 6.5) [31] for 1 or 7 days. For the Repair group, Curodont™ Repair was mixed with 50 µL of distilled water, applied on the enamel surfaces, and left undisturbed for 5 min. After that, excess was removed with absorbing paper, and 50 µL of Ca^2+^ and PO_4_ supersaturated solution was applied for 1 min. Excess was removed with gauze, and the specimens were stored in artificial saliva for 1 or 7 days. For the Protect group, Curodont™ Protect was applied on the enamel surfaces and left to react for 5 min. Excess was removed with absorbing paper, and 50 µL of Ca^2+^ and PO_4_ supersaturated solution was applied for 1 min. After that, excess was removed, and specimens were stored as preciously described for the Repair group. The specimens from all groups were immersed in artificial saliva at 37 °C and tested for Knoop microhardness and surface roughness after 1 and 7 days.

The Knoop microhardness (KNH) analysis was performed before tooth bleaching (baseline) and 24 h and 7 days after the bleaching treatments. Knoop microhardness analysis was conducted using a micro-indenter (HMV 2000, Shimadzu, Tokyo, Japan), with a static load of 50 gF, for 10 s. Three (3) equidistant (approximately 500 µm) indentations were made on each specimen and the arithmetic average was calculated.

Mean surface roughness (Ra) was evaluated using a roughness tester device (Surfcorder SE 1700, Kosaka, Tokyo, Japan). Three (3) readings per specimen were performed in different directions from equidistant points on the enamel surfaces. Readings were standardized with a 0.25 mm cutoff, 1 mm/s speed, and a 0.01 µm to 8 µm minimum/maximum tolerance range. The arithmetic average of the three readings was obtained in Ra. Roughness readings were performed before bleaching (baseline) and 24 h and 7 days after bleaching treatments.

After microhardness and surface roughness tests, the specimens were coated with gold/palladium (SCD 050; Balzer, Schaan, Liechtenstein) and analyzed in a scanning electron microscope (SEM) (JSM-5600LV; Jeol, Tokyo Japan) operated at 15 kV with a working distance of 10 mm and a spot size of 25 and a magnification of 2500×. The SEM analysis was performed to enable a visual and qualitative comparison of the enamel surfaces after the different treatments. To compare the treated surfaces with sound enamel, three (3) other specimens were fabricated using the same protocol described above without being subjected to bleaching.

Data normality was checked by the Shapiro–Wilk test and the homoscedasticity of variances by Levene’s test. Data comparisons were performed using repeated measures ANOVA and the Bonferroni post hoc test, with two independent variables: treatment and time was considered as a sub-parcel. Dunnett’s test was performed to compare the baseline with the specimens 24 h and 7 days post-bleaching treatment. For all statistical analyses, a 95% level of significance (α = 0.05) and power analysis of 80% was used.

## 3. Results

### 3.1. Knoop Microhardness (KHN)

Knoop microhardness (KHN) values are presented in Table 3 and Figure 4. The repeated measure ANOVA test indicated a significant interaction between the time elapsed and the treatment performed in relation to KHN (*p* = 0.001). After 24 h post-bleaching, there was no significant difference among groups. However, after 7 days, the highest KHN values were found for Protect and Repair. There was no significant difference between the negative control (bleached with no treatment) and NaF groups, which presented significantly lower KHN values than Repair and Protect.

To verify surface hardness over time, Dunnett’s test was used to compare the baseline with other storage times for all groups (Table 3). The Protect group results were significantly higher at 24 h (*p* = 0.037) and 7 days (*p* = 0.017) than the baseline KHN values. At 7 days, the Repair group values were considerably higher than the baseline values (*p* = 0.027). There was no difference between the negative control and NaF and the baseline values. The Protect (11.02% at 24 h; 16.7% at 7 days) and Repair (16.02% at 24 h; 21.10% at 7 days) groups showed higher and constant increases in KHN. Control and NaF KHN values decreased initially (8.16% and 7.25%, respectively) and experienced a slight increase at 7 days in the saliva solution (1.7% and 7.4%, respectively), as shown in Figure 4.

### 3.2. Surface Roughness (Ra)

Surface roughness (Ra) values are presented in Table 4 and Figure 5. There was no interaction between study factors (*p* = 0.097). Overall, there was a difference between the evaluation times. Values for the baseline (*p* = 0.000) were significantly lower than those at 24 h (*p* = 0.000), which were significantly higher than those at 7 days (*p* = 0.031). To verify the recovery Ra for all treated groups, a Dunnett’s test was performed (*p* < 0.05). The results showed a significant increase in Ra when compared to the negative control (*p* = 0.000) and Protect (*p* = 0.019) groups, which means that the roughness of the enamel surfaces was significantly higher than those of the baseline. The Repair group maintained surface roughness at both times. After 7 days in saliva solution, all groups recovered the original Ra, similar to sound teeth.

### 3.3. Scanning Electron Microscopy (SEM)

SEM images are presented in Figure 6. SEM demonstrated different morphological aspects after 7 days regarding the treatments. The unbleached enamel surface (Figure 6A) presented a slightly rough aspect and enamel pores. The enamel surface for the negative control (Figure 6B) evidenced either depressions or dissolution of the enamel surface. Subtle irregularities and slight parallel grooves were noticed on the surfaces treated with NaF (Figure 6C). In contrast, when treated with self-assembly peptide-containing materials (Repair and Protect), the enamel surfaces seemed smoother and defects caused by bleaching were less evident (Figure 6D,E).

## 4. Discussion

Erosive damage and mineral loss induced by bleaching agents remain critical concerns related to intrinsic and extrinsic factors associated with tooth structure [27]. In addition, using fluoridated materials to treat these affected surfaces might have some drawbacks once the fluoridated phase formed on enamel is no longer stable [7]. In this study, we evaluated the effect of self-assembling peptide-based materials (Curodont™) on the microhardness and surface roughness of bovine enamel after bleaching with 35% hydrogen peroxide. Materials containing the self-assembling peptide P_11_-4 significantly increased enamel microhardness when compared with other groups after 7 days post-bleaching. Thus, the first hypothesis of this study was accepted.

P_11_-4 contained in the Repair and Protect treatments is connected with interesting interactions between enamel structure and the minerals present in saliva. This interaction could explain the effects of the Repair and Protect treatments on microhardness after 7 days when compared with NaF. Initially, P_11_-4 is an α-peptide that in low pH conditions self-assembles into β-sheet amyloids (protein secondary structure). Due to the peptide backbone pattern of hydrogen bonds between amino hydrogens and carboxyl oxygen atoms, β-sheet are formed, which allows for a 3D-matrix [32]. P_11_-4 has a hydrogel appearance at low pHs due to β-sheet-forming domains that promote toughness and strength, similar to what is observed in muscle tissues, silk, and amyloid fibrils [32,33]. When applied to enamel surfaces, this self-assembling peptide diffuses through the pores of the mineralized tissue to reach the subsurface [34]. Then, the peptide monomers spontaneously form a three-dimensional matrix through hydrogen-bonding and mimic the enamel matrix to serve as a scaffold that attracts calcium and phosphate ions [27,35]. As a result, P_11_-4 can fill up cracks and/or eroded areas of enamel, preventing or efficiently treating sub-surface carious lesions.

Hydroxyapatite deposition can reduce tooth hypersensitivity caused by either bleaching or exposition of dentin tubules upon gingival recession [35]. It has been reported that after every application of Curodont™ Repair or Curodont™ Protect an apatite-like tissue of 10 µm–50 µm can form on enamel [35,36]. Therefore, the mechanism of action of P_11_-4 consists of a gradual formation and possible progressive crystal maturation [37]. Apart from that, at seven days, the Repair and Protect specimens presented similar results at all timepoints and were different from those of the NaF group. Sodium fluoride reacts with hydroxyl (OH) groups present in hydroxyapatite and forms fluorapatite [14], which is considered a mechanism of mineral deposition on tooth structures. However, the KHN values for NaF did not change after seven days, which can be associated with the formation of a non-stable crystalline phase [38]. Since Curodont™ Protect also has sodium fluoride in its composition and its values increased after seven days, it is possible to associate the microhardness results with the self-assembling peptide. In this way, despite the importance of fluoride for the demineralization and remineralization process [14], P_11_-4-containing materials were more efficient in recovering microhardness.

Moreover, Curodont™ Repair and NaF were able to reestablish initial enamel surface roughness in 24 h, whereas for the other treatments similar results were only observed after seven days. Thus, the second hypothesis was partially rejected, since Curodont™ Protect was not able to recover surface roughness immediately but only after seven days. Increase in enamel surface roughness occurs because hydrogen peroxide has a pH of 5.0, which is below the demineralization limit of the dental enamel and causes mineral loss [38]. Furthermore, for at-home dental bleaching, carbamide peroxide releases 50% of its hydrogen peroxide content in the first two hours, which can be sustained for up to ten hours [39]. This sustained release of peroxides can lead to enamel erosion and consequently increased surface roughness and dental permeability [11].

In theory, after a few days, dental enamel returns to its initial state through the buffering capacity of saliva [7]. However, this is still controversial because teeth bleached in vivo with 35% carbamide peroxide lost the aprismatic enamel layer and damage was not repaired after 90 days [40]. Thus, previous studies recommend the use of 2% NaF (9000 ppm) as a complementary therapy to accelerate the remineralization process [41,42,43].

Compared to the Protect group, the quick recovery in surface roughness observed for the Repair group could be explained by the viscosity and diffusion mechanisms in play [16,20] and the fast transition from a Newtonian fluid to a nematic gel in low pH conditions [35]. Curodont™ Protect is a remineralizing gel, while Curodont™ Repair is a lyophilized peptide that turns into an aqueous solution when in contact with water, with a high ability of spreading on a hard surface. For this reason, it is expected that Curodont™ Protect should take more time to interpenetrate the subsurface and recover enamel structure. It is also hypothesized that the incorporation of other components, such as remineralizing agents, flavorings, and thickeners, could interfere with the peptide’s mechanism of action, slowing the reaction. Furthermore, the residues of those substances may remain on the surface, which can increase surface roughness.

Although after seven days all groups recovered initial roughness values, SEM demonstrated that groups not treated with P_11_-4-based materials presented some irregular areas. Qualitatively, it was possible to observe a higher uniformity of enamel morphology in the Protect and Repair groups. Such images corroborate the SEM micrographs from studies regarding early carious lesion treatments using self-assembling peptides [16,44]. Further, calcium phosphates such as hydroxyapatite have whitening properties independently of bleaching or polishing [45,46]. The Protect and/or Repair products recruit ions to form hydroxyapatite, which could change the color of enamel or even improve bleaching; however, this was not in the scope of this study, and should be further investigated. Furthermore, mineral gain, which could more accurately confirm the effects of the peptide on dental remineralization, was not evaluated, and further studies should be carried out involving tooth sensitivity and the structuring of enamel.

Important and preliminary findings have been presented in this study and it is possible to suggest the use of P_11_-4-containing materials as viable alternatives to treat bleached enamel surfaces and even decrease post-bleaching tooth sensitivity. In a clinical situation, the clinician could apply the materials after each bleaching stage.

## 5. Conclusions

Commercially available products containing self-assembly peptide P_11_-4 used as a post-bleaching treatment were able to recover surface roughness and increase the microhardness of bleached enamel as compared with sound enamel.

## Figures and Tables

**Figure 1 jfb-13-00079-f001:**
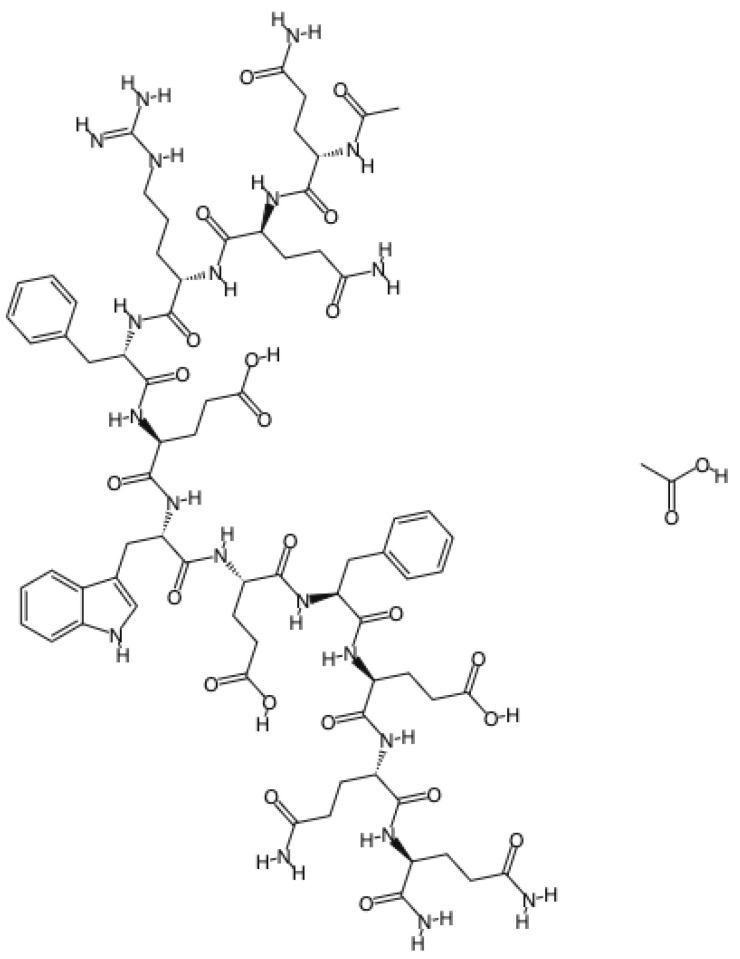
Chemical structure of P_11_-4.

**Figure 2 jfb-13-00079-f002:**
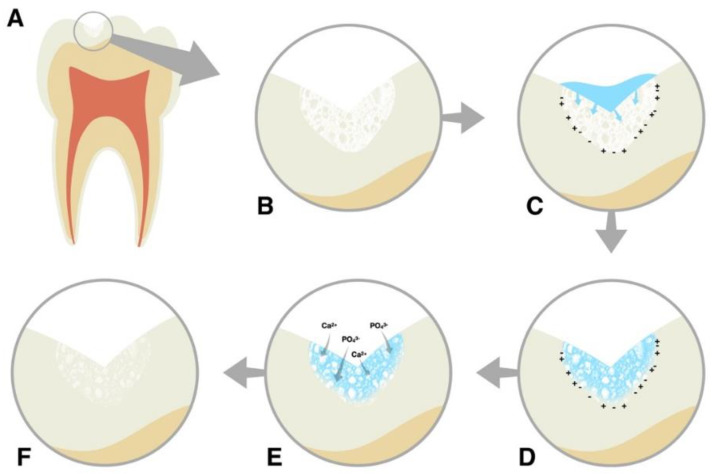
Mechanism of action of P_11_-4. (**A**,**B**) Initial caries lesion characterized by demineralized and porous enamel (white spot lesion). (**C**) The monomeric solution of P_11_-4 on the initial caries lesion surface diffuses through the pores of demineralized enamel. (**D**) Triggered by local pH on the enamel surface, P_11_-4 undergoes self-assembly by β-sheet formation; thus, a 3D matrix is formed in the lesion. (**E**) A 3D matrix with a high affinity for Ca^2+^ and PO_4_^3−^ and nucleation of minerals forms until maturation in HAP-like crystals (de novo mineralization). (**F**) Biomimetic mineralized enamel.

**Figure 3 jfb-13-00079-f003:**
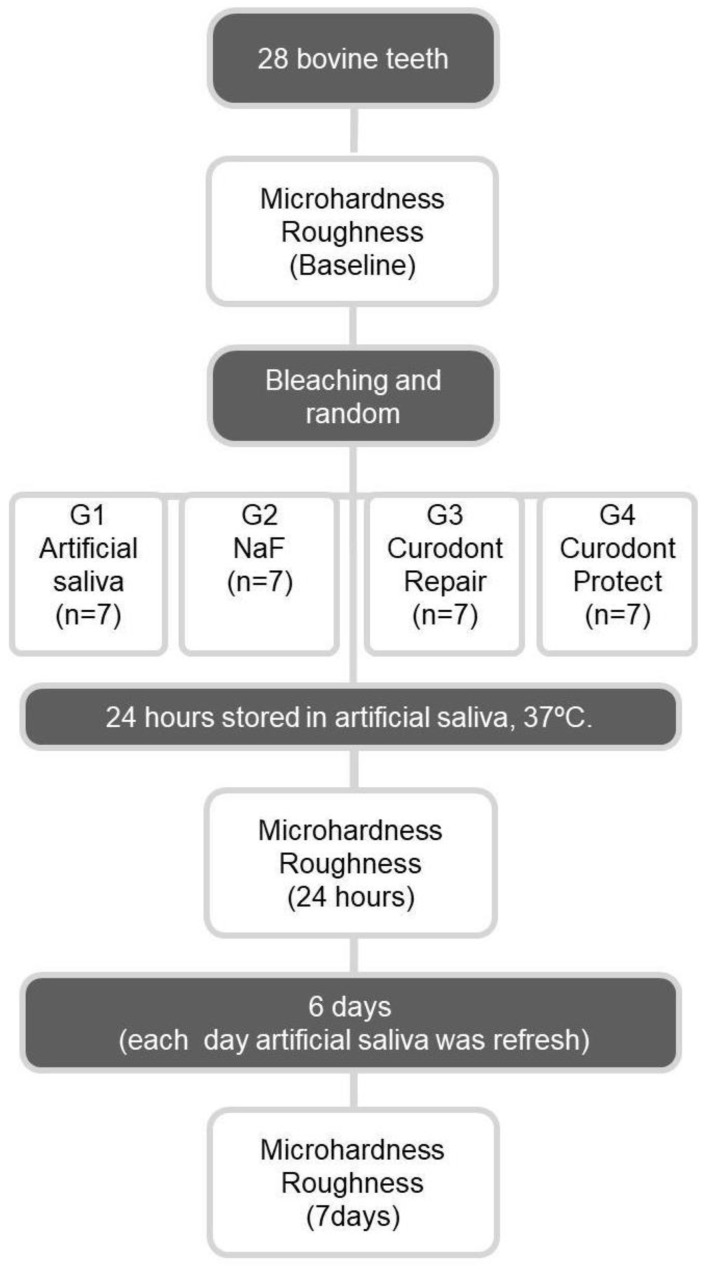
Study design flowchart.

**Figure 4 jfb-13-00079-f004:**
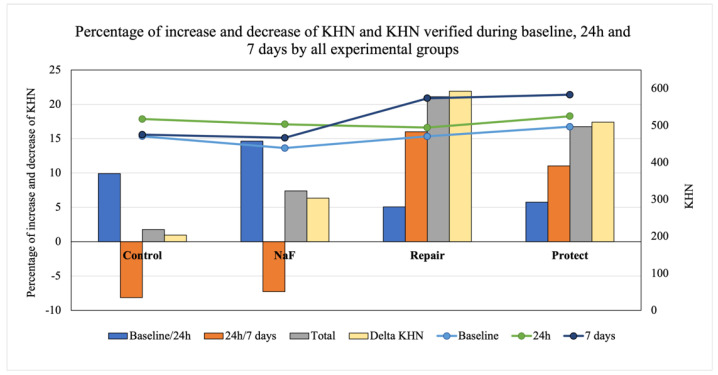
Analysis of microhardness (Knoop hardness—KHN) of the bovine enamel subjected to bleaching and treated with each material over time. Bars represent the percentage increases and decreases in KHN and KHN verified at baseline and at 24 h and 7 days after treatment for all experimental groups. Gray bars represent microhardness gain after the time allowed to elapse for each treatment. Yellow bars represent the percentages of variations considering initial and final KHN values measured at different times according to each material. Rows show KHN raw values for each group. Blue row—KHN at baseline; green row—KHN at 24 h; dark-blue row—KHN at 7 days.

**Figure 5 jfb-13-00079-f005:**
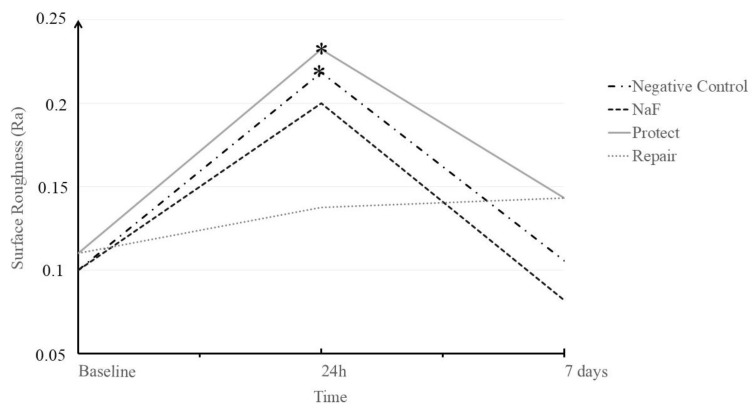
Analysis of surface roughness (Ra) of the bovine enamel subjected to bleaching and treated with each material over time. * Significant differences between the time allowed to elapse for each treatment group.

**Figure 6 jfb-13-00079-f006:**
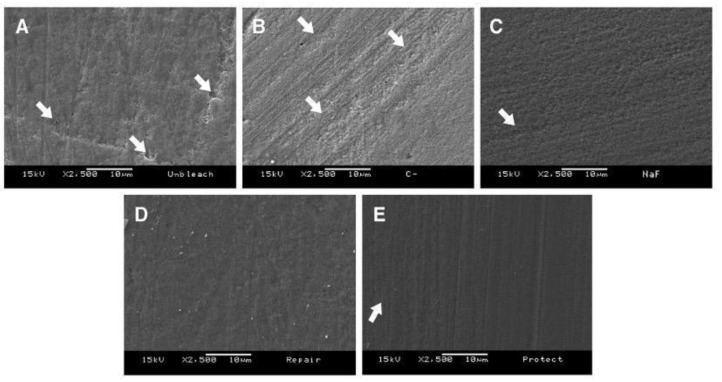
SEM (operated at 15 kV, with a WD of 10 mm, a spot size of 25, and a magnification of 2500×) micrographs of the surface specimens from each group. (**A**) Micrograph of the unbleached bovine enamel specimen. White arrows indicate slight rugosity and enamel pores. (**B**) Micrograph of the bleached, untreated negative control (C-) enamel specimen. White arrows indicate some depressions in or dissolution of the enamel surface. (**C**) Micrograph of the specimen surface-treated with NaF. White arrow indicates subtle irregularities and slight parallel grooves. (**D**) Micrograph of the specimen surface-treated with Curodont™ Repair. (**E**) Micrograph of the specimen surface-treated with Curodont™ Protect. The enamel surface seems smoother and the defects caused by bleaching were less evident (**D**,**E**). The white arrow (**E**) shows parallel lines on the surface resulting from polishing.

**Table 1 jfb-13-00079-t001:** Groups and surface treatment protocols.

Groups	Treatment Protocols
C-	35% Carbamide Peroxide, with storage in artificial saliva
NaF	35% Carbamide Peroxide, 2% NaF 9.000 ppm, supersaturated Ca^2+^ and PO_4_ solution, with storage in artificial saliva
Repair	35% Carbamide Peroxide, Curodont™ Repair, supersaturated Ca^2+^ and PO_4_ solution, with storage in artificial saliva for 1 or 7 days
Protect	35% Carbamide Peroxide, Curodont™ Protect, supersaturated Ca^2+^ and PO_4_ solution, with storage in artificial saliva

**Table 2 jfb-13-00079-t002:** Materials’ compositions and application times.

Material (Manufacturers)	Composition	Application Time
Whiteness Hp Maxx 35% (FGM, Joinville, SC, Brazil)	Hydrogen Peroxide 35%, thickener, red dye, glycol, and water	3 × 15 min
Flugel (Nova DFL, Rio de Janeiro, RJ, Brazil	2% NaF, 9000 ppm	1 min
Curodont™ Repair (Credentis AG, Dorfstrasse, Windisch, Switzerland)	Peptide P_11_-4 (amino acid sequence: Ace-Gln-Gln-Arg-Phe-Glu-Trp-Glu-Phe-Glu-Gln-Gln-NH_2_)	5 min
Curodont™ Protect (Credentis AG, Dorfstrasse, Windisch, Switzerland)	Hydrogenated Starch Hydrolysate, Aqua, Hydrated Silica, PEG-8, Cellulose Gum, Sodium Monofluorophosphate, Aroma, Sodium Saccharin, Citric Acid, Sodium Hydroxide, Dicalcium Phosphate, Oligopeptide-104, Calcium Glycerophosphate, Sodium Chloride, Sodium Sulfate, Limonene, Cinnamal, CI 42090	5 min
Ca^2+^ and PO_4_^3-^ solution	Saturated solution of Ca^2+^ and PO_4_^3−^ (1.5 mmol/L calcium, 0.9 mmol/L phosphate, and 150 mol/L KCl in 20 mmol/L cacodylic buffer, pH 7.0)	1 min
Artificial saliva	1.5 mM CaCl_2_, 0.9 mM KH_2_PO_4_, 130 mM KCl, and 20 mM Hepes, pH 6.5	Stored for 24 h and 7 days

**Table 3 jfb-13-00079-t003:** Means and standard deviations (SDs) of KHN values measured at different times according to each material.

Groups	24 h	7 Days
Negative control	517.44 (46.41) Aa	475.22 (58.95) Ba
NaF	503.00 (37.30) Aa	465.50 (41.50) Ba
Repair	494.33 (28.94) Ab	572.50 (79.04) Aa *
Protect	525.17 (51.58) Ab *	583.00 (74.76) Aa *

Capital letters represent significant differences between time periods (row); lowercase letters represent significant differences between treatment groups (column). * Represents differences between the baseline and other storage times for all groups as determined by Dunnett’s test.

**Table 4 jfb-13-00079-t004:** Means and standard deviations (SDs) of surface roughness (Ra) values measured at the different times for each material.

Groups	Baseline	24 h	7 days	Average
Negative control	0.122 (0.054)	0.221 (0.059)	0.117 (0.033)	0.222 (0.074)
NaF	0.117 (0.059)	0.205 (0.068)	0.168 (0.169)	0.163 (0.044)
Repair	0.118 (0.020)	0.133 (0.035)	0.140 (0.039)	0.126 (0.010)
Protect	0.123 (0.029)	0.236 (0.068)	0.141 (0.085)	0.189 (0.067)
Average	0.120 (0.041) B	0.199 (0.069) A	0.142 (0.094) B	

Different capital letters in the same row indicate statistical differences among storage times.

## Data Availability

Not applicable.

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
