# Peer review of "Effect of a Self-Assembly Peptide on Surface Roughness and Hardness of Bleached Enamel"

_jfb, 2022, doi:10.3390/jfb13020079_

Round 1

Reviewer 1 Report

This manuscript is very informative aimed to he aim of this preliminary study was to evaluate the efficacy of different self-assembling peptide-containing materials, as a post-bleaching procedure, on recovering surface properties of enamel. The method of the  survey is described in a clear, informative and repeatable manner; therefore, the manuscript is valuable, well structured, and well written. 
The topic of the manuscript is relevant, and I recommend publication. 

Author Response

Dear reviewer #1,

The authors thank you for your kind comments. 

Sincerely, 

The authors

Reviewer 2 Report

In this manuscript, Puppin-Rontani and the co-authors present how the self-assembly of oligopeptide P11-4 affects the surface roughness and hardness of bleached enamel. They use proper methods to realise the hypothesis and the results are convincing for me. Herein, the reviewer suggests this manuscript can be accepted as is.

Author Response

Dear reviewer #2,

Thank you for your comments about our study. The authors greatly appreciate it.

Sincerely,

The authors

Reviewer 3 Report

Tha autors reported about the positive effect of a self-assembling peptide (P11-4) on the enamel surface protection after post-bleaching treatments. The paper is a compartive study with and without fluoride, and sodium fluoride (NaF 2%) on Knoop microhardness (KHN) and Roughness (Ra–μm) of bleached enamel with in-office bleaching regimen on recovering
the sound enamel properties.

The manuscript has a scientific soundness. In my opinion the interest can be low. However, the paper is well described and exposed. For this, I suggest a minor revision step according to the following points:

1) Please, include the chemical structure of P11-4.

2) Add additional details for the secondary structure, methodology of formulation and rheological feature of the matrix.

3)Other peptide-based self-assembling systems are studied for dental application. For examples, mixed matrices were proposed for tissue engineering applications. Report some examples( e.g.: 10.1039/d0sm00825g;  10.1021/acs.biomac.7b00876)

4) Please, provide a picture for the mechanism of action of the peptide.

Author Response

Dear Reviewer #3,

Thank you for all your comments. The authors greatly appreciate your hard work in improving the quality of the manuscript. We addressed bellow each of the comments individually. Please, find our responses (in red) to your comments (in blue).

Sincerely,

Regina Maria Puppin-Rontani, DDS, MSc, PhD

“Moderate English changes required Tha autors reported about the positive effect of a self-assembling peptide (P11-4) on the enamel surface protection after post-bleaching treatments. The paper is a compartive study with and without fluoride, and sodium fluoride (NaF 2%) on Knoop microhardness (KHN) and Roughness (Ra–μm) of bleached enamel with in-office bleaching regimen on recovering the sound enamel properties.

The manuscript has a scientific soundness. In my opinion the interest can be low. However, the paper is well described and exposed. For this, I suggest a minor revision step according to the following points:

Thank you for your comments. Indeed, the manuscript has been completely revised by a native speaker in terms of English language (grammar, consistency, spell-check, etc). Please, find all comments and responses below.

1)Please, include the chemical structure of P11-4.

Thank you for your comment. The authors agree that it is rather important to represent the chemical structure of P11-4 in our study. Chemical structure depiction of P11-4 was made using the nomenclature Ac-Gln-Gln-Arg-Phe-Glu-Trp-Glu-Phe-Glu-Gln-Gln-NH2.CH3CO2H according to the International Union of Pure and Applied Chemistry (IUPAC) on Molview.org. (website).

(see structural formula image).

Figure 1. Chemical structure of P11-4. (Please, see the figure at the file attached)

2) Add additional details for the secondary structure, methodology of formulation, and rheological feature of the matrix.

Thank you for your comments. Authors believed this specific information were important and added the following to the manuscript: (lines 280-286): “Initially, P11-4 is an α-peptide that in low pH self-assembles into β-sheet amyloids, which refers to protein secondary structure. Due to the peptide backbone pattern of hydrogen bonds between the amino hydrogen and carboxyl oxygen atoms β-sheet are formed, which allows for a 3D-matrix[32]. P11-4 has a hydrogel appearance at low pHs, due to β-sheet forming domains that promote toughness and strength, similarly observed in muscle tissues, silk, and amyloid fibrils[32, 33].”

3)Other peptide-based self-assembling systems are studied for dental application. For examples, mixed matrices were proposed for tissue engineering applications. Report some examples (e.g.: 10.1039/d0sm00825g; 10.1021/acs.biomac.7b00876)

Thank you for your suggestion. The following has been added to the manuscript (lines 84-92): “Other peptide-based self-assembling systems have been studied for biomedical applications such as PEG8-(FY)3. It is a hybrid polymer–peptide conjugate that can self-assemble into a self-supporting soft hydrogel over dry and wet surfaces. In dentistry, its use could be important as well, since it acts as a scaffold and supports cell growth[24]. An Arginine-containing Peptide Hydrogel, enriched with Hydroxyapatite, seems to be promising for tooth mineralization. This multicomponent peptide-based hydrogel is composed of fluorenyl-9-methoxycarbonyl diphenylalanine (FmocFF), which provides rigidity and stability to the hydrogel, and Fmoc-arginine (FmocR), which mediates high affinity to hydroxyapatite (HAP), due to arginine moiety[25].

4) Please, provide a picture for the mechanism of action of the peptide”

Thank you for this suggestion. Indeed, a representation of the mechanism of action was crucial for the readers. Authors created the following figure to illustrate this process. Please, find the Figure 2 at the file attached. Thank you.

Figure 2. A and B – Initial caries lesion characterized by demineralized and porous enamel (white spot lesion); C – The monomeric solution of P11-4 on the initial caries lesion surface and diffuses through the pores of demineralized enamel; D – Triggered by local pH on the enamel surface, P11-4 undergoes self-assembly by β-sheet formation, thus a 3D-matrix is formed into the lesion; E – 3D-matrix is a high affinity for Ca2+ and PO43- and nucleation of minerals occurs until maturation in HAP-like crystals (De novo mineralization); F – Biomimetic mineralized enamel.

Reviewer 4 Report

The present study aimed to compare the effect of the SAP P11-4 with and without fluoride, and sodium fluoride on surface characteristics of enamel (Knoop microhardness, roughness) after bleaching.

Some comments:

1. General: there are different P11-4 products available (repair, protect etc.) which have different indications and different recommendation for use. This should be addressed clearly in the manuscript.

2. The company has changed (it´s not credentis anymore but vVardis, Switzerland).

3. Abstract: please report numerical data such as p-values and other relevant values in the results so that the abstract can summarize the main findings including data (it´s not enough only writing: higher…, lowest… etc.).

4. Introdcution, page 2, line 80: there are newer studies available which report the remineralisation capability of SAP which should be referred to.

5. Material and Methods: there is no sample size calculation reported. What was the basis of using n = 7 specimen in each group? What is the power of the results then?

6. Page 9, line 304: Curodont protect is rather a gel than toothpaste.

Author Response

Dear Reviewer #4,

Thank you for all your comments. The authors greatly appreciate your hard work in improving the quality of the manuscript. We addressed bellow each of the comments individually. Please, find our responses (in italic) to your comments (in normal).

Sincerely,

Regina Maria Puppin-Rontani, DDS, MSc, PhD

“The present study aimed to compare the effect of the SAP P11-4 with and without fluoride, and sodium fluoride on surface characteristics of enamel (Knoop microhardness, roughness) after bleaching.

Some comments:

  1. General: there are different P11-4 products available (repair, protect etc.) which have different indications and different recommendation for use. This should be addressed clearly in the manuscript.

Thank you, reviewer is absolutely correct. It makes perfect sense. The following information has been added to the manuscript (lines 72-75): “This peptide can be found in Curodont™ Repair, a commercial product that is indicated to initial caries lesions, and in Curodont™ Protect, a professional remineralizer gel indicated during orthodontic treatments and after bleaching”. Thank you again.

  1. The company has changed (it´s not credentis anymore but vVardis, Switzerland).

Thank you for this comment. Authors searched for this information and in fact, Credentis has become the research and development department within vVardis. However, the products were sent to us by Credentis and used as Credentis information for using them. Besides that, Curodont™ Repair and Curodont™ Protect were used following the manufacture’s instruction.

  1. Abstract: please report numerical data such as p-values and other relevant values in the results so that the abstract can summarize the main findings including data (it´s not enough only writing: higher…, lowest… etc.).

Thank you for this comment, authors agreed with this observation. Information has been added on paper (line 25-30): “There was a significant interaction between study factors (p=0.001). After 7 days, Repair (572.50 ± 79.04) and Protect (583.00 ± 74.76) increased KHN surface beyond baseline values, higher when compared to NaF (465.50 ± 41.50) and C- (475.22 ± 58.95). There was no significant difference in KHN at 24h among groups (p=0.587). At 24h after bleaching, Repair was significant different from all groups (p<0.05). Repair showed the lowest Ra (μm) values (0.133 ± 0.035).”

  1. Introdcution, page 2, line 80: there are newer studies available which report the remineralisation capability of SAP which should be referred to.

Thank you for this comment, and authors absolutely agree. We included relevant clinical trials in this area, as follows:

  • Bröseler, F.; Tietmann, C.; Bommer, C.; Drechsel, T.; Heinzel-Gutenbrunner, M.; Jepsen, S. Randomised clinical trial investigating self-assembling peptide P(11)-4 in the treatment of early caries. Clin Oral Investig 2020, 24, 123-132, doi:10.1007/s00784-019-02901-4.
  • Doberdoli, D.; Bommer, C.; Begzati, A.; Haliti, F.; Heinzel-Gutenbrunner, M.; Juric, H. Randomized Clinical Trial investigating Self-Assembling Peptide P(11)-4 for Treatment of Early Occlusal Caries. Sci Rep 2020, 10, 4195, doi:10.1038/s41598-020-60815-8.
  • Kobeissi, R.; Badr, S.B.; Osman, E. Effectiveness of Self-assembling Peptide P(11)-4 Compared to Tricalcium Phosphate Fluoride Varnish in Remineralization of White Spot Lesions: A Clinical Randomized Trial. Int J Clin Pediatr Dent 2020, 13, 451-456, doi:10.5005/jp-journals-10005-1804.
  • Sedlakova Kondelova, P.; Mannaa, A.; Bommer, C.; Abdelaziz, M.; Daeniker, L.; di Bella, E.; Krejci, I. Efficacy of P(11)-4 for the treatment of initial buccal caries: a randomized clinical trial. Sci Rep 2020, 10, 20211, doi:10.1038/s41598-020-77057-3.
  • Welk, A.; Ratzmann, A.; Reich, M.; Krey, K.F.; Schwahn, C. Effect of self-assembling peptide P(11)-4 on orthodontic treatment-induced carious lesions. Sci Rep 2020, 10, 6819, doi:10.1038/s41598-020-63633-0.

  1. Material and Methods: there is no sample size calculation reported. What was the basis of using n = 7 specimen in each group? What is the power of the results then?

Thank you for pointing out this issue. Actually, we did a sample calculation by conducting a pilot study previously. However, after many updates, we oversaw this such important information. It has been added to the manuscript: (lines 120-122) “Sample calculation was accomplished based on a previously conducted pilot study considering a=0.05 and b=0.20. Thus, seven specimens per group was deemed adequate.”; and power analysis information (lines 196-197) “For all statistical analyses, it was considered a 95% level of significance (α=0.05) and power analysis of 80%.”.

  1. Page 9, line 304: Curodont protect is rather a gel than toothpaste.

Authors thank you for this comment. We agreed and made a correction. Please find more information added to the manuscript (lines 324-328): “Curodont™ Protect is a remineralizer gel while Curodont™ Repair is a lyophilized peptide, that turns into an aqueous solution when in contact with water, with high ability of spreading on a hard surface. For this reason, it is expected that Curodont™ Protect should take more time to interpenetrate the subsurface and recover enamel structure”.

Round 2

Reviewer 4 Report

The authors revised the manuscript according to the reviewer´s suggestions and the manuscript can be accepted.

However, an important paper dealing with the P11-4 has not been cited:

Randomised in situ clinical trial investigating self-assembling peptide matrix P11-4 in the prevention of artificial caries lesions. DOI:10.1038/s41598-018-36536-4

This is of more relevance here than the current reference 19 which should be replaced.